# Mechanical, Dynamic-Mechanical, Thermal and Decomposition Behavior of 3D-Printed PLA Reinforced with CaCO_3_ Fillers from Natural Resources

**DOI:** 10.3390/polym14132646

**Published:** 2022-06-29

**Authors:** Cristina Pavon, Miguel Aldas, María Dolores Samper, Dana Luca Motoc, Santiago Ferrandiz, Juan López-Martínez

**Affiliations:** 1Instituto de Tecnología de Materiales (ITM), Universitat Politècnica de València (UPV), 03801 Alcoy, Spain; crisppavonv@gmail.com (C.P.); masammad@upvnet.upv.es (M.D.S.); sferrand@mcm.upv.es (S.F.); jlopezm@mcm.upv.es (J.L.-M.); 2Departamento de Ciencia de Alimentos y Biotecnología, Facultad de Ingeniería Química y Agroindustria, Escuela Politécnica Nacional, 170517 Quito, Ecuador; miguel.aldas@epn.edu.ec; 3Faculty of Mechanical Engineering, Transilvania University of Brasov (UniTBv), 500036 Brasov, Romania

**Keywords:** 3D-printing, PLA polymer, biological filler, mechanical, dynamic-mechanical, thermal, decomposition, wettability, surface appearance

## Abstract

This study evaluates the effect of CaCO_3_ fillers extracted from waste eggshells on 3D-printed PLA performance. Samples of neat PLA and PLA reinforced with CaCO_3_ fillers embedded with different wt.% were prepared using an FDM (fused deposition modeling) technology. The samples were examined using mechanical, dynamic mechanical, thermal, and thermal decomposition analyses. The results revealed increasing elastic moduli, tensile strength, and flexure as the filler content increased. The rheological results from the MFR tests showed that the filler content did not influence the PLA-based samples’ processability. Further, the thermal degradation of neat and various CaCO_3_-wt.%-reinforced PLA specimens revealed relatively small discrepancies in their exposure to the temperature increase, mainly concerning the eggshell organic components and volatile components, from their processability up to 300 °C. By contrast, the increased filler content positively shifted the peaks along the temperature scale at the maximum degradation rate. Additionally, the weight content of the natural reinforcement strongly influenced the surface wettability and appearance of the samples. Further, the SEM analysis featured both the presence of interlayer disturbances and the interfacial compatibility the PLA with the selected fillers.

## 1. Introduction

Additive manufacturing (AM), or 3-dimensional printing (3D printing), is a technology that translates a digitized solid model into physical models without cutting or casting machines [1]. The physical object is formed through the combination of 2-D cross-sections of a finite thickness in a layer-by-layer addition sequence [2,3]. In this way, AM allows the production of complex-shape objects or multiple-component objects in a shorter time and at low costs compared to the traditional manufacturing process [4]. Additionally, AM maximizes savings on raw materials during the process [5].

Additive manufacturing allows the processing of a wide range of materials, including metals, polymers, ceramics, and concrete [6]. In particular, polymers receive significant attention due to their ease of production, availability, and low cost [2,7]. AM processing uses polymers in diverse forms, such as reactive, liquid solutions, and thermoplastic melts [2]. Furthermore, they can be processed in a significant number of methods. The most frequently used techniques are stereolithography (SLA), fused deposition modelling (FDM), selective laser melting (SLM), and selective laser sintering (SLS). SLA is a technique that requires the raw material in a liquid form. By contrast, the other techniques use a solid material [3,5].

Stratasys. Inc. developed FDM in the USA in 1990 for processing traditional thermoplastics [8,9]. FDM is one of the most commonly used techniques throughout the world. FDM is a low-cost, easy technique [4,5,10], and can produce complex geometrical parts neatly and safely in an office-friendly environment [8]. The FDM process involves heating the polymer in a solid-state (e.g., filament) at the nozzle to reach a semi-liquid state and then extruding and depositing it on the platform or into the previous layers. When the nozzle deposits the whole layer, the platform moves down by the height of one layer and begins to print the next layer [3]. The nozzle temperature must be adapted to the material’s melting point. Heating the deposition platform (build-plate) reduces thermal shrinkage after deposition [2].

The most common thermoplastics processed by FDM are poly(lactic acid) (PLA), polystyrene (PS), acrylonitrile butadiene styrene (ABS), polyamide (PA), polyethylene terephthalate (PET) [4,10], polycarbonate (PC), and PC-ABS blends [8]. One of the most favored thermoplastics for FDM 3-D printing is PLA [10] because of its relatively low melting point (150–160 °C), which implies energy savings [11]. Moreover, PLA is considered a user-friendly material that can be processed without producing toxic fumes [12,13]. PLA is a biodegradable and compostable polymer produced via fermentation or chemical synthesis from the bio-derived monomer, lactic acid (2-hydroxy propionic acid) [14].

PLA’s stereochemical structure can easily be modified by controlling the ratio between the L- and D-isomers to yield high-molecular-weight amorphous or crystalline polymers [14]. The stereochemistry and thermal history of PLA directly influence its crystallinity and, therefore, its properties in general [14,15]. As a result, lactic-acid-based polymers present different grades with a wide variety of mechanical properties, ranging from soft and elastic plastics to stiff and high-strength materials [10,15]. Semi-crystalline PLA provides higher mechanical properties. It has an approximate tensile modulus of 3 GPa, a tensile strength of 50–70 MPa, a flexural modulus of 5 GPa, a flexural strength of 100 MPa, and an elongation at break of about 4% [16]. However, PLA is rigid and brittle at room temperature due to its T_g_ (~60 °C) [15], and it has some issues regarding its low thermal stability, high degradation rate during processing, and drawability [10,13]. Drawability and processability are essential in FDM technology because they influence feedstock filament production and layer deposition during printing [10]. To enhance its processability, thermal stability, physical properties, or/and appearance for specific applications and reducing its costs, PLA is combined with (1) inorganic fillers as calcium carbonate nanofillers [10], hydroxyapatite [17], (2) fibers as sheep’s wool [18], or (3) polysaccharides as starch [19].

In this context, the present work aims to determine the material properties (e.g., mechanical, dynamic-mechanical, thermal, decomposition, etc.), surface wettability, and appearance of 3D-printed neat PLA thermoplastics and those reinforced with various wt.% (e.g., 3 and 5 wt.%) calcium carbonate (CaCO_3_) fillers. The main aim is to improve the thermal stability, overcome problems related to the mechanical properties of these 3D-printed specimens, and support the enlargement of their application area, as other material properties seem appropriate. Additionally, this study brings forth a deployment of calcium carbonate fillers acquired directly from natural sources, namely eggshells, considered a residue in the industry.

## 2. Materials and Methods

### 2.1. Materials

Commercial poly(lactic acid) (PLA) pellets Ingeo Biopolymer 2003D were acquired from NatureWorks LLC. (Minnetonka, MN, USA) and deployed as the polymer matrix. Poly(lactic acid) was selected, accounting for its general purpose, high molecular weight, transparency, ease of processing, and wide range of potential applications. Table 1 lists the physical and mechanical properties of the pellets as provided by the supplier. Eggshell was the source for the calcium carbonate (CaCO_3_) selected as filler. For this purpose, the eggshells were dried for 8 h at 90 °C, mechanically triturated, and sieved to 400 microns as mesh size [20]. No additional purification step of the eggshell as CaCO_3_ primary source applied. The natural fillers were embedded with the neat PLA matrix in 3 and 5 wt.% and further referred to as PLA-0% CaCO_3_, PLA-3% CaCO_3_, and PLA-5% CaCO_3_, respectively.

### 2.2. Filament Production and Temperature Printing Parameters

The PLA–CaCO_3_ formulations were manually premixed in plastic containers and conditioned at 25 °C and 50 ± 5% relative humidity to produce the filament required for the 3D-printing process. The materials were processed in a co-rotating twin-screw extruder Dupra S.L (Castalla, Spain) at 50 rpm, using the following temperature profile from hopper to die: 160, 170, 180, and 190 °C, respectively. Next, the resulting material was milled into pellets. Finally, the pellets were processed in a filament extruder, model EX2, from Filabot (Barre, PA, USA) at 190 °C using a 2.95-millimeter nozzle diameter. Additionally, neat PLA pellets were identically processed to enable further property comparison. The resulting filaments’ diameters varied between 2.80 and 3.00 mm.

### 2.3. Manufacturing of the Test Specimens

A 3D printer BCN3D (Barcelona, Spain) manufactured the test specimens. The nozzle temperature and build-plate temperature were selected based upon the results from differential scanning calorimetry (DSC) run on all PLA-based batches. The parameter used to print the test specimens were: nozzle diameter, 0.6 mm; infill density, 20%; raster angle, 45°; number of extruders, 1; layer height, 0.1 mm; printing speed, 20 mm/s; number of layers, 10; printing bed, glass. Figure 1 represents the geometry and dimensions of the printed test specimens associated with the *n* and *n* + 1 layer, respectively.

### 2.4. Tensile and Flexural Testing

Tensile and flexural tests were performed according to ISO 527-1:2012 and ISO 178:2019, respectively [22,23]. Testing devices LS100 and LR5K Plus from Lloyd Instruments (Bognor Regis, UK) running in tensile and 3-point bending modes, respectively, were operated to retrieve the mechanical properties of specimens. The tensile tests use a gauge length of 30 mm and 1 mm/min crosshead rate. At the same time, a 10-kilonewton load cell was operated to retrieve flexural properties of the specimens under a bending rate of 10 mm/min and length span of 64 mm. Five specimens of each formulation were selected and characterized for tensile and flexural strength, respectively, and this manuscript reports their mean and standard deviation. The significant differences were assessed at 95% confidence level according to Tukey’s test using a one-way analysis of variance (ANOVA).

### 2.5. Differential Scanning Calorimetry (DSC)

Differential scanning calorimetry (DSC) tests determined the nozzle and glass-plate temperatures for 3D printing. The DSC analyses were performed on a DSC 821 calorimeter from Mettler-Toledo (Schwerzenbach, Switzerland) under a controlled nitrogen atmosphere (flow rate 30 mL/min). PLA-based samples (5–6 mg) were subjected to the following thermal cycle: dynamic heating from 25 °C to 150 °C, to remove thermal history from the printing process, followed by cooling to −50 °C, and then further dynamic heating up to 250 °C. The peak of the DSC curve of PLA-based specimens made it possible to identify the melting temperature (T_m_).

### 2.6. Melt Flow Index (MFI)

The rheology characterization was performed through melt mass-flow-rate (MFR) measurement, following ISO 1133-1:2005 standard, using a Metrotec GmbH brand plastometer model Ars Faar, with 2.16 kg at 190 °C. The MFR value reported corresponds to the mean value of eight measurements.

### 2.7. Thermogravimetric Analysis (TGA)

Thermogravimetric analysis (TGA) was conducted on a TGA PT1000 device from Linseis (Selb, Germany), weighing 15 and 20 mg samples. Samples were heated under TGA dynamic mode at 10 °C/min, from 35 °C to 700 °C, under a controlled nitrogen atmosphere (flow rate 30 mL/min). The onset (T_5%_) and endset (T_95%_) degradation temperatures were retrieved for 5% and 95% sample mass loss, respectively. Further, the maximum degradation rate temperature (T_max_) was identified from the minimum of the TGA curve first derivative (DTG).

### 2.8. Dynamic Mechanical Thermal Analysis (DMTA)

Dynamic-mechanical thermal analysis (DMTA) was performed on a TA Instrument AR G2 rheometer (New Castle, DE, USA). Samples were exposed to a temperature sweep from 35 °C to 140 °C with a heating rate of 2 °C/min, an oscillation frequency of 1 Hz, and 0.1% of maximum deformation. The samples were prismatic specimens with dimensions of 40 ×10 × 4 mm^3^ (L × l × h).

### 2.9. Microstructural Characterization

Field emission scanning electron microscopy (FESEM) from the fracture surface of the impact specimens was carried out on a Carl Zeiss AG Ultra 55 microscope (Oberkochen, Germany) at 1 kV, displaying a resolution of 1.7 nm. Prior to investigations, each sample was sputter-coated with a thin layer of gold to avoid electrostatic charging under the electron beam with of a Sputter Mod Coater Emitech SC7620 from Quorum Technologies (East Sussex, UK).

### 2.10. Surface Characterization

Surface-color evaluation was performed on a Colorflex-Diff2 458/08 colorimeter from HunterLab (Reston, VA, USA) under the CIE L*a*b* color space. The L*, a*, and b* coordinates and the yellowness index (YI) were reported. Furthermore, the total color difference (∆E) was calculated using Equation (1) [24]:(1)$$\Delta E=\sqrt{\Delta a^{2}+\Delta b^{2}+\Delta L^{2}}$$

Five individual measurements were performed on each specimen, and this manuscript reports their average and standard deviation.

An optical goniometer, EasyDrop-FM140 from Kruss Equipments (Hamburg, Germany), was used to assess samples’ wettability and roughness. Images were processed using the Drop Shape Analysis software. Two values for water contact angles were obtained, accounting for two different surfaces, one in direct contact with the build plate and the other corresponding to the up-side surface. The measurements were carried out at room temperature using distilled water.

Further comparison with the color formulation and WCA of reference (PLA) was carried out by one-way ANOVA using Tukey’s test, with 95% significant differences in the confidence level.

## 3. Results and Discussion

### 3.1. Differential Scanning Calorimetry and Melt-Flow Index

Figure 2 shows the differential scanning calorimetry curves of the PLA and PLA–CaCO_3_ formulations. The CaCO_3_ constitutive did not change the glass transition temperature values (T_g_) irrespective of its wt.% (3% or 5%, respectively). This result is in accordance with the results reported by Nekhamanurak et al. (2012) [25]. Furthermore, the melting (T_m_) and the endset (T_end_) temperatures of the formulations were identical to those of the neat PLA sample.

Furthermore, there were no significant differences in the PLA-0%, PLA-3%, and PLA-5% melt mass-flow rate (MFR), displaying a mean value of 10.28 ± 0.57. These results indicate that the PLA-reinforced composites exhibited identical rheological characteristics, strengthening the idea that irrespective of CaCO_3_ wt.%, the latter did not influence the PLA matrix processability. Consequently, the materials were processed under identical temperature conditions. Table 2 presents the T_g_, T_m_, T_end_, and MFR values.

All the formulations selected the printing temperature above the end melting temperature. The extrusion temperature of the 3D printing process is generally higher than that of filament extrusion (190 °C) [26] because of the short residence time of the material in the 3-D printer chamber and the reduced contribution of the shear stress generated by the loading gear [13,26]. Based on prior trials, different nozzle and build-in plate temperatures for the first and the other printed layers were imposed, as follows: first layer, 215 °C and 55 °C; bulk-printed layers, 210 °C and 40 °C, respectively. The temperatures selected for the first layer were slightly higher than those of the following bulk layers to allow PLA and PLA-based composites to melt better and adhere to the built-in plate. Figure 3 shows the printed samples. As can be seen, all the printed surfaces were uniform irrespective of the reinforcement wt.%, with minor differences on the 5% CaCO_3_ specimen.

The thermo-plasticity of the polymer filament is an essential property of this method, which allows the filaments to fuse during printing and solidify at room temperature after printing. The layer thickness, width, and orientation of filaments and the air gap (in the same layer or between layers) are the main processing parameters that affect the mechanical properties of printed parts [6,8].

### 3.2. Tensile, Flexural, and Impact Testing

The results of the mechanical characterization of the neat PLA and PLA composites herein are listed in Table 3. The Young’s moduli from both the tensile and the flexural test revealed an increasing tendency with the addition of CaCO_3_ fillers, as expected, and followed the rule of mixture (RoM). Correspondingly the values retrieved from the static mechanical tests differed significantly between 28% to 42% in the tensile and about 1.50% to 2.25% in the flexure test. These discrepancies were due to the filament material composition and morphology, the manufacturing technology, and, less probably, to the testing procedure and equipment [14,17].

Concerning the filler distribution within the PLA matrix, the morphological studies from the literature revealed an increasing trend for the elongation at break based upon the improved interaction with the matrix [27,28]. The results reported by Nekhamanurak et al. (2012) on their CaCO_3_-reinforced PLA composites were similar with herein findings [25]. Further, Nekhamanurak et al. found that a 5 wt.% fraction of embedded nano-fillers decreases the elongation at break and acts as a stress concentrator, since fillers may tend to agglomerate [25,29].

Compared with the mechanical properties of raw pellets, the tensile and flexural properties of samples such as those reported above are strongly influenced by the manufacturing technology. In support of this argument, Silva et al. (2007) showed that the mechanical properties of polymers experience a decrease of 30% when processed by FDM technology compared to those obtained by deploying a compression-molding technique [30]. Further, the infill degree contributes to the degradation of mechanical properties, as shown by Silva et al. and Ahmed et al. [30,31,32]. According to Ahmed et al. (2019)’s studies on commercial PLA filaments, the Young’s modulus of 3-D printed PLA at an infill level of 20% and a printing angle of 45° retrieved from tensile tests was about 889.33 ± 49.34 MPa [31]. These values closely match our values from the flexural tests. Nevertheless, the filaments described herein were produced under a laboratory-controlled environment, and their retrieved mechanical properties were influenced by the factors described above.

Yao et al. (2019) underlined that the flexural properties of 3D-printed parts depend mainly on the design parameters and printing settings [5]. Further, it is widely acknowledged by thermoplastic composite manufacturers that the addition of CaCO_3_ fillers within the polymer resin contributes to inner-stress release during the manufacturing process [10]. According to the mechanical tests, the direct consequences of the factors mentioned above can be observed in the decreasing trend of the tensile and flexural strength. The tensile strength decreased from 6% to 25%, whereas the flexural strength was roughly 5% and 10% relative to the neat PLA values.

Further, the relative toughness of the neat and CaCO_3_-wt.%-reinforced PLA-based 3D-printed samples experienced a decrease in line with the increase in the natural filler content, to about 18% and 20%, respectively. Consequently, the increase in the wt.% of this natural filler significantly affected the toughness of 3D-printed PLA-reinforced specimens.

### 3.3. Thermograviemtric Analysis

Figure 4 presents the weight loss of the PLA-0%, PLA-3%, and PLA-5% CaCO_3_ as a function of temperature. The first step reveals the loss of moisture, greasy, and other volatile compounds within the fillers. We further referred to the temperature range up to 300 °C as the first weight-loss interval. Thus, 2% of the total weight was lost within the 30–120 °C temperature range, mainly due to volatile compounds from the manufacturing process of the samples [33]. The filler volume fraction seems to have influenced the weight loss as the temperature increased. Thus, within the 150–300 °C temperature range, the PLA-3% CaCO_3_ exhibited an extra 1% weight loss, while the PLA-5% CaCO_3_ was about 3%. This supplementary weight loss can be related to the decomposition of organic material [34] within the eggshell composition [35].

The second step in the neat and PLA-reinforced excerpt degradation revealed the association between the highest weight loss and the PLA chain scission, between 300 and 400 °C [36]. In the PLA with CaCO_3_, this degradation step started at about 320 °C, while for the PLA–CaCO_3_ formulations, it began around 340 °C. A shift towards higher temperatures was determined through the peak values for the degradation rates, accounting for the presence of the CaCO_3_ fillers [33,37]. Finally, the last degradation step revealed a slightly higher endset degradation temperature (T_95%_) for the PLA-reinforced CaCO_3_ formulations. Table 4 lists the main temperature values of the degradation process (e.g., T_max_ and T_95%_). The tendencies encountered in these variations and the retrieved values may enable us to present a statement regarding the filler’s effect upon the thermal stability of the PLA-reinforced excerpts. Thus, the addition of CaCO_3_ enabled improvements in thermal stability that contrasted with the results reported by Nekhamanurak et al. (2014) [36]. These discrepancies may have been related to the particle agglomeration which, in our case, did not influence the properties under discussion [36].

### 3.4. Dynamic-Mechanical Analysis

Figure 5 shows the storage moduli (G′) and loss factor (Tan δ) of the neat and PLA-reinforced formulations. The DMA revealed two transitions and three different states (e.g., glassy, leathery, and rubbery) for the neat and PLA-reinforced composites. Table 5 presents the DMA parameters. The glassy region was relatively narrow, covering the 35–60 °C temperature range. The glass transition temperatures (T_g_) of the neat PLA, 3% and 5-wt.%-CaCO_3_-reinforced PLA, retrieved as the maximum of the Tan(δ) curve, were about 60.9 °C, 61.2 °C, and 60.8 °C, respectively. These values were consistent with those from the DSC analysis. Further, in the rubbery plateau, as the temperatures exceeded 120 °C, the storage moduli increased for all the specimens. This increase suggests an increase in the stiffness of the materials, which can be attributed to the cold crystallization of the PLA structure [38,39,40]. Moreover, adding CaCO_3_ shifted the temperature at the beginning of the cold crystallization to lower values, suggesting a nucleating effect. Furthermore, in the temperature range beyond 100 °C, the storage moduli (G′) of the PLA–CaCO_3_ formulations were lower than those of the neat PLA. These discrepancies in the storage module can be ascribed to the rearrangement into crystallites of the PLA polymer chains above the glass transition (T_g_) [41,42]. Therefore, the addition of the CaCO_3_ fillers interrupted the rearrangement, leading to an increase in the elastic behavior of the PLA-based composites.

### 3.5. Microstructural Characterization

Figure 6 presents the study samples’ scanning electron microscopy (FESEM) results. This analysis assessed the final 3D-printed samples. In Figure 6a, on the left side, it is possible to note the fragile fracture section of the PLA. In Figure 6b and Figure 6c, the 3 wt.% and 5 wt.% CaCO_3_ materials, respectively, are shown. In both cases, the presence of the natural filler (e.g., eggshells) is evident, since the surface of the materials is not as smooth as the PLA sample. In both cases, the samples’ layers are not pasted, contrary to the shape detected in the PLA sample. The lack of adherence is explained by the presence of the CaCO_3_ fillers, which affect the printing process of these formulations. Moreover, in Figure 6c, it is possible to observe the presence of porosity due to the filler content. The defects in the structure suggest a non-optimal adhesion between the CaCO_3_ and PLA matrix [43]. These defects in the microstructure of the formulations led to the decrease in the mechanical properties (e.g., tensile strength and elongation at break) discussed above.

### 3.6. Surface Characterization

Table 6 presents the water contact angle (WCA) and the color coordinates in the CIEL*a*b* space of the neat PLA, PLA-3%, and PLA-5%. Regarding the water contact angle, adding the CaCO_3_ fillers significantly reduced (*p* < 0.05) the WCA of the PLA-based composites on both surfaces, free and build-plate. The WCA reduced about 17% for the 3 wt.% CaCO_3_ reinforced PLA on the build-plate surface and 24% for the 5 wt.% specimen. By contrast, on the free surface, the WCA experienced a 3% reduction in the PLA-3% and about 12% in the PLA-5%. The decrease in the WCA was attributed to the hydrophilic sites on the surface of the CaCO_3_ fillers [44], and, to a greater extent, to an increase in the superficial roughness of the specimens [10]. Moreover, the differences between the WCA values reflect the roughness differences from one surface to the other. As already acknowledged, free surfaces exhibited higher roughness than the build-plate surfaces because the former does not have a mechanism to contains or shape its surface.

The color measurement results show that incorporating CaCO_3_ fillers increases the brightness of PLA (*p* < 0.05) significantly. Adding 3 wt.% CaCO_3_ increases the lightness by about 13 points, while adding 5 wt.% CaCO_3_ increases it by about 19 points. This result means that the formulations obtained a whiter coloration due to the inherent coloration of the CaCO_3_ fillers. The a* coefficient shows that the samples exhibited a green hue that decreased significantly (*p* < 0.05) with the increase in the CaCO_3_ percentage [45]. As a* approaches zero, the samples do not reveal coloration [18]. The b* coordinates revealed the yellow hue exhibited by the samples. The retrieved values of the yellow index (YI) increased significantly (*p* < 0.05) with the CaCO_3_ wt.% content [45]. Finally, the total difference in color (∆E) shows statistical differences between the samples (*p* < 0.05), with values that differ by more than two units. Therefore, the color change is appreciable to the human eye [46,47].

## 4. Conclusions

This paper presents the results of tensile, flexural, and impact tests on 3D-printed neat PLA and PLA reinforced with CaCO_3_ fillers from natural resources (e.g., eggshells) and their thermal and decomposition behavior. In addition, the samples’ wettability and appearance were measured to aid in the characterization of the different PLA-based and wt.%-CaCO_3_-reinforced composites.

The results revealed a decreasing tendency in the tensile (e.g., 6% and 25%) and flexural strength (e.g., 5% and 10%) concerning the neat PLA, performed under a controlled loading mode. These values diminish the application area’s attractiveness; the material selection criteria may include specifications as to the direction of the strength variation. On the other hand, the elastic moduli, both tensile and flexural, increased with the increase in the wt.% of the fillers (e.g., tensile, 28% and 42%, respectively; flexural—1.5% and 2.5%) compared with the neat PLA and following the rule of mixture (RoM).

Depending on material properties, such as dynamic-mechanical, thermal, or decomposition, the results revealed less or no variations due to the changes in the wt.% content of the CaCO_3_ fillers. Thus, the glass transition and melting temperatures of the neat and wt.%-CaCO_3_-reinforced PLA composites retrieved from the DSC runs showed no differences. The decomposition of the PLA–CaCO_3_ composites experienced identical weight loss tendencies, developed in three steps that were identified according to different temperature ranges. Up to 120 °C, there were no differences in the recorded weight loss for the neat and PLA-based composites related to the filler contents. Between 150 and 300 °C, the weight loss increased by about 1% and 3% of the total size. The increase in filler content accounts for the decomposition of the organic components with the eggshell. The maximum decomposition rate occurred around 370 °C, with a positive shift in the peaks sized on the temperature scale for the PLA-reinforced specimens. The dynamic-mechanical analysis provided results that were consistent with the above findings. The glass transition temperatures, as retrieved for the peaks of the loss factor, were in line with the results from the DSC tests, about 60 °C for all the PLA and PLA-based composites. Furthermore, the loss tangent peak decreased with the increase in the content of the CaCO_3_ filler, without broadening its curve envelope over the temperature range to a greater extent. The SEM analysis revealed interlayer disturbances of the PLA matrix by the fillers, and an increasingly porous appearance of the deposited layers.

As a result of this research, 3D-printed PLA reinforced with calcium carbonate (CaCO_3_) from eggshells, under different wt.%, can be considered a viable alternative to other natural reinforcements as its mechanical, dynamic-mechanical, thermal, and decomposition properties prove its competitiveness.

## Figures and Tables

**Figure 1 polymers-14-02646-f001:**
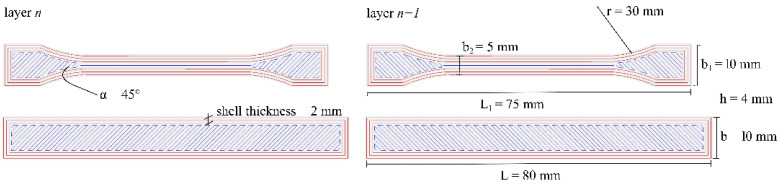
Dog-bone and prismatic test specimen dimensions and infill orientation pattern.

**Figure 2 polymers-14-02646-f002:**
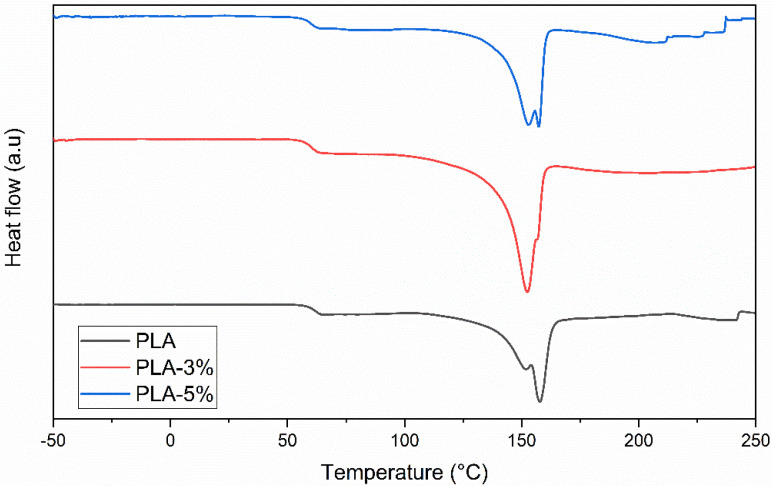
Differential scanning calorimetry curves of PLA-0%, PLA-3%, and PLA-5%CaCO_3._

**Figure 3 polymers-14-02646-f003:**
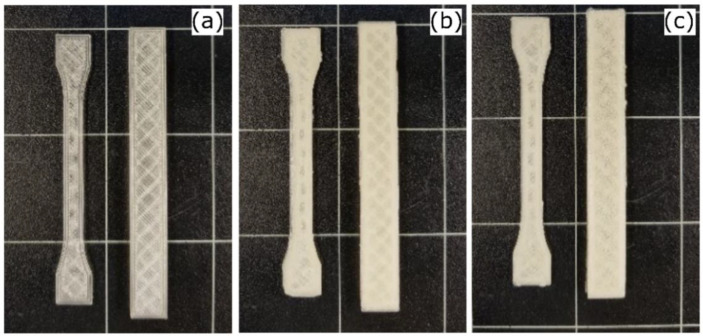
Test specimens of (**a**) PLA-0%, (**b**) PLA-3%, and (**c**) PLA-5%CaCO_3._

**Figure 4 polymers-14-02646-f004:**
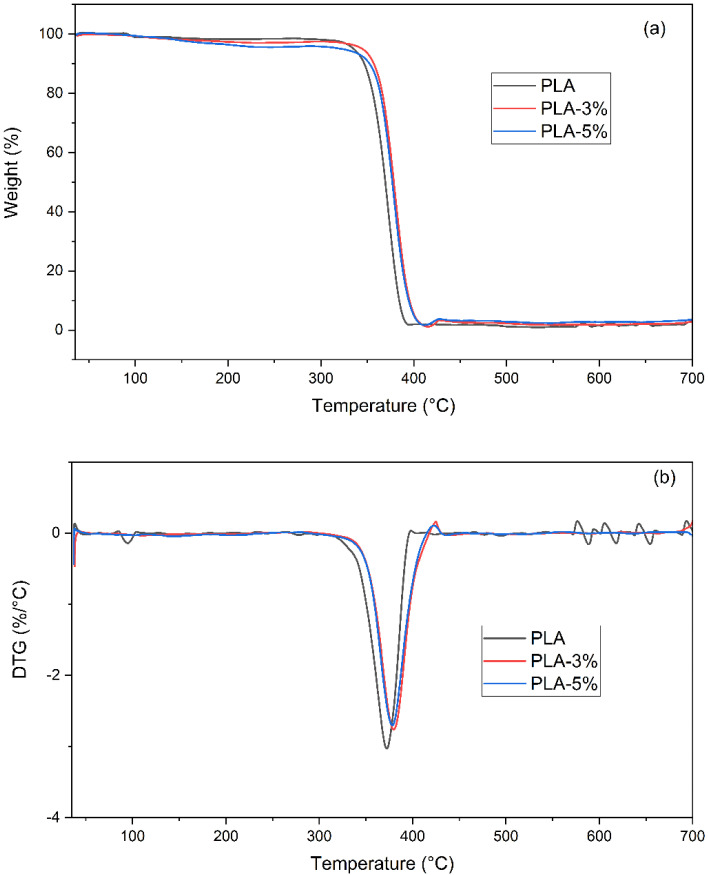
(**a**) TGA and (**b**) DTG of PLA, PLA-3%, and PLA-5% CaCO_3_ fillers.

**Figure 5 polymers-14-02646-f005:**
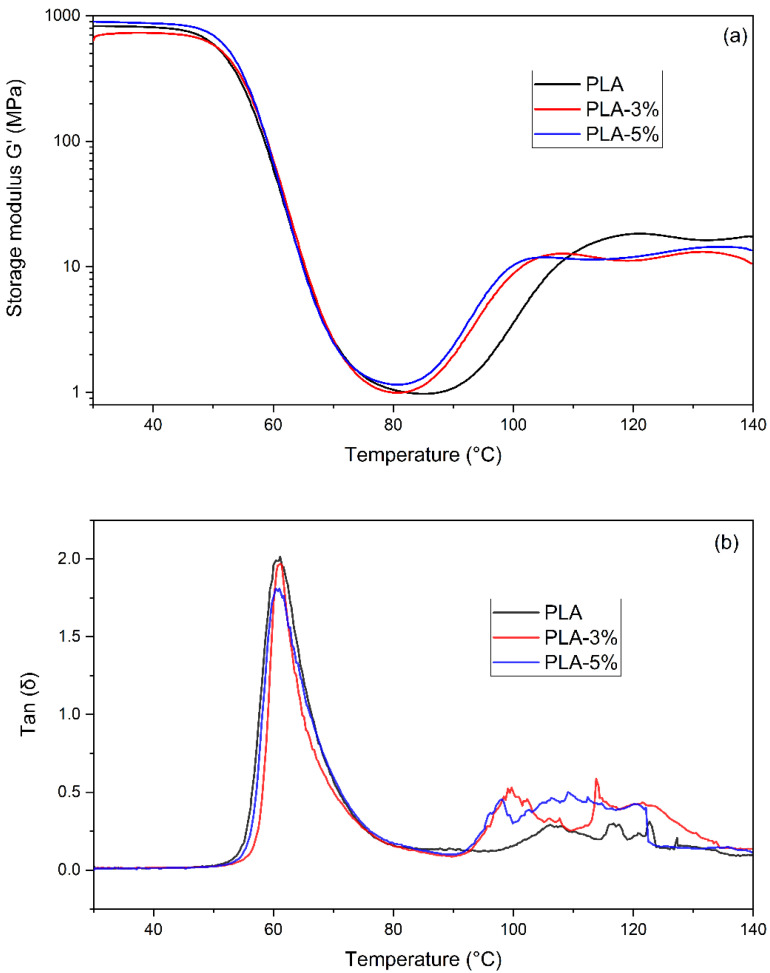
(**a**) Storage modulus and (**b**) tan(δ) variation as a function of temperature of PLA, PLA-3%, and PLA-5% CaCO_3_ fillers.

**Figure 6 polymers-14-02646-f006:**
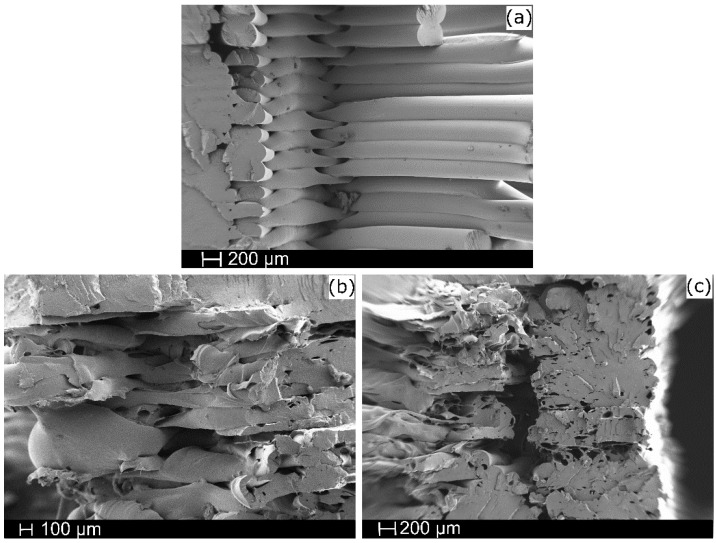
SEM microstructural characterization of (**a**) neat PLA, (**b**) PLA-3%, and (**c**) PLA-5%.

**Table 1 polymers-14-02646-t001:** The physical and mechanical properties of as-received PLA Ingeo 2003D pellets *.

Property	Value	Property	Value
Density	1.24 g/cm^3^	Izod impact	0.16 J/cm
Tensile strength	53.0 MPa	Heat distortion temperature	55 °C
Flexural strength	82.7 MPa	Glass transition temperature	55–60 °C
Tensile modulus	3.61 GPa	Melting temperature	140–160 °C
Flexural modulus	3.83 GPa	Transmission (visible)	90%

* Ingeo™ biopolymer 2003D technical data sheet [21].

**Table 2 polymers-14-02646-t002:** Glass transition temperature, melting temperature, endset temperature, and melt mass-flow rate of neat PLA, PLA-3%, and PLA-5% CaCO_3_.

Specimen	T_g_ (°C)	T_m_ (°C)	T_end_ (°C)	MFR (g/10 min at 190 °C)
PLA-0%	60	158	185	10.10 ± 0.60
PLA-3%	60	152	183	10.43 ± 0.40
PLA-5%	60	157	186	10.32 ± 0.72

**Table 3 polymers-14-02646-t003:** Tensile, flexural, and impact-resistance properties and area of tensile stress–strain curve (Toughness) of 3-D printed neat PLA, PLA-3%, and PLA-5%CaCO_3_.

	Tensile Properties			Flexural Properties		
Material	Young’s Modulus (MPa)	Tensile Strength (MPa)	Elongation at Break (%)	Young’s Modulus in Flexure (MPa)	Flexural Strength (MPa)	Charpy’s Impact Energy (kJ/m^2^)
PLA-0%	1201.2 ± 95.76	42.04 ± 1.98	9.79 ± 1.57	805.64 ± 10.113	45.53 ± 4.70	7.75 ± 0.13 ^a^
PLA-3%	1534.6 ± 185.71	39.26 ± 2.98	4.26 ± 0.63	817.89 ±21.54	43.21 ± 5.65	6.33 ± 0.91 ^b^
PLA-5%	1708.1 ± 43.22	31.22 ± 1.28	4.32 ± 0.45	823.43 ± 14.98	40.59 ± 6.67	6.19 ± 0.11 ^b^

^a,b^ Different letters within the same property show statistically significant differences between formulations (*p* < 0.05).

**Table 4 polymers-14-02646-t004:** Onset degradation temperature (T5%), temperature of maximum degradation rate (Tmax), and endset degradation temperature (T95%) for PLA and the formulations with CaCO_3_.

Material	T_5%_ (°C)	T_max_ (°C)	T_95%_ (°C)
PLA-0%	335	372	387
PLA-3%	334	379	400
PLA-5%	321	380	401

**Table 5 polymers-14-02646-t005:** Storage modulus before glass transition and crystallization of PLA, PLA-3%, and PLA-5% CaCO_3_ fillers, and temperature at Tan δ peak.

Material	G′ at 40 °C (MPa)	G′ at 80 °C (MPa)	G′ at 120 °C (MPa)	T at Tan δ Peak
PLA-0%	824	1.0	18	60.9
PLA-3%	721	1.1	12	61.2
PLA-5%	890	1.2	12	60.8

**Table 6 polymers-14-02646-t006:** Water contact angle (WCA) and color parameters for the CIEL*a*b* space for PLA-0%, PLA-3%, and PLA-5% CaCO_3_ fillers.

Material	WCA Build-Plate Surface	WCA Free Surface	L*	a*	b*	YI	ΔE
PLA-0%	54.5 ± 2.3 ^a^	81.1 ± 2.3 ^a^	59.6 ± 1.5 ^a^	−0.9 ± 0.1 ^a^	3.2 ± 0.2 ^a^	8.5 ± 0.7 ^a^	0.7 ± 0.3 ^a^
PLA-3%	44.9 ± 2.8 ^b^	77.9 ± 2.5 ^b^	72.1 ± 1.4 ^b^	−0.6 ± 0.1 ^b^	8.7 ± 0.4 ^b^	20.6 ± 0.9 ^b^	13.4 ± 1.0 ^b^
PLA-5%	41.5 ± 1.8 ^c^	71.2 ± 2.0 ^c^	78.3 ± 0.5 ^c^	−0.2 ± 0.1 ^c^	12.6 ± 0.3 ^c^	27.9 ± 0.6 ^c^	21.6 ± 0.7 ^c^

^a–c^ Different letters within the same property show statistically significant differences between formulations (*p* < 0.05).

## Data Availability

The data presented in this study are available upon request from the corresponding author.

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
