# Peer review of "Mechanical, Dynamic-Mechanical, Thermal and Decomposition Behavior of 3D-Printed PLA Reinforced with CaCO3 Fillers from Natural Resources"

_polymers, 2022, doi:10.3390/polym14132646_

Round 1
Reviewer 1 Report
In general, the proposed paper is quite well written. However, some improvements can be made.
1. Check Eng Language and constructions of the senetences
2. Add the sources of the given data, for example, Tables 1, 3, etc
3. Some references are quite old and maybe not necessary (1995, 1996), as well as that based on the Google origin.
After improvements be done, the paper can be considered for next step
Reviewer 2 Report
Comments to polymers-1788081
If the CaCO3 fillers could simultaneously increase the elastic moduli, tensile, and flexure, it would be a very interesting technology.
Tensile, flexural and impact resistance properties and area of tensile stress–strain Curve 244 (Toughness) of 3-D printed neat PLA, PLA-3% and PLA-5%CaCO3 are listed in Table 3. However, the authors did not compare the significant difference between groups (p<0.05). Water contact angle (WCA) and colour parameters listed in Table 6. The difference between groups are derived from random error?
Round 2
Reviewer 2 Report
significantly increases the brightness of PLA (p < 0.05).
'(p < 0.05) ' should be followed with the word 'significantly'
